# Intestinal Cells-on-Chip for Permeability Studies

**DOI:** 10.3390/mi15121464

**Published:** 2024-11-30

**Authors:** Marit Keuper-Navis, Hossein Eslami Amirabadi, Joanne Donkers, Markus Walles, Birk Poller, Bo Heming, Lisanne Pieters, Bjorn de Wagenaar, Adam Myszczyszyn, Theo Sinnige, Bart Spee, Rosalinde Masereeuw, Evita van de Steeg

**Affiliations:** 1Department of Metabolic Health Research, Netherlands Organisation for Applied Scientific Research (TNO), 2333 BE Leiden, The Netherlands; 2AZAR Innovations, 3584 CH Utrecht, The Netherlands; 3ARTIC Technologies B.V., 5612 AE Eindhoven, The Netherlands; 4Pharmacokinetic Sciences, Biomedical Research, Novartis Pharma AG, CH-4056 Basel, Switzerland; 5TNO Holst Centre, 5656 AE Eindhoven, The Netherlands; 6Department of Clinical Sciences, Faculty of Veterinary Medicine, Utrecht University, 3584 CT Utrecht, The Netherlands; 7Institute for Risk Assessment Sciences (IRAS), Utrecht University, 3584 CT Utrecht, The Netherlands; 8Division of Pharmacology, Utrecht Institute for Pharmaceutical Sciences (UIPS), Utrecht University, 3584 CG Utrecht, The Netherlands

**Keywords:** intestine-on-chip, cell monolayer, in vitro model, intestinal absorption, intestinal barrier, permeability

## Abstract

Background: To accurately measure permeability of compounds in the intestine, there is a need for preclinical in vitro models that accurately represent the specificity, integrity and complexity of the human small intestinal barrier. Intestine-on-chip systems hold considerable promise as testing platforms, but several characteristics still require optimization and further development. Methods: An established intestine-on-chip model for tissue explants was adopted for intestinal cell monolayer culture. A 3D-printed culture disc was designed to allow cell culture in static conditions and subsequent permeability studies in a dynamic environment. Membrane characteristics and standardized read-outs were investigated and compared to traditional permeability studies under static conditions. Results: By starting cultures outside the chip in conventional wells plates, the new cell disc design could support accurate cell monolayer formation for both Caco-2 and human enteroids. When transferred to the chip with laminar flow, there was accurate detection of barrier integrity (FD4 and Cascade Blue) and permeability (atenolol/antipyrine). Both flow and membrane characteristics had a significant impact on permeability outcomes. Conclusions: This novel intestinal cell-on-chip system offers large flexibility for intestinal permeability studies, although it still requires validation with more compounds to reveal its full potential.

## 1. Introduction

The small intestine serves as a major barrier separating the internal body from the external environment. The intestinal epithelium specifically allows selective transport of key biomolecules, while preventing harmful substances from entering the body. In addition, compounds can be metabolized in the intestinal wall before entering the body, as part of the first-pass metabolism. Accurate intestinal barrier integrity and functionality are therefore pivotal for whole body homeostasis [1,2,3].

Building a preclinical in vitro model with similar properties to the human intestinal tissue is essential for various applications, including studying selective permeability. The model should accurately represent the (1) specificity, (2) integrity and (3) complexity of the intestinal barrier in vivo, which is still lacking in conventional cell culture models. Advantages and disadvantages of current existing intestinal in vitro and ex vivo models are extensively discussed elsewhere [4,5,6]. The overall conclusion is that, in particular, the new complex in vitro models, i.e., intestine-on-chip systems, hold considerable promise as preclinical testing platforms. These systems are able to mimic key aspects of human physiology by combining recent advances in tissue engineering and microfabrication [7]. The chips are designed to recapitulate the in vivo microenvironment of the cells, including chemical and mechanical stimuli, thereby maintaining tissue-specific functions [8]. Pimenta et al. provided a comprehensive overview of the main features and components of intestine-on-chip models reported in the literature [9]. The large variation in fabrication techniques, materials used, architectural scaffold, fluid flow, mechanical stimulation and the different applications of the models makes comparisons between studies challenging.

An example of an intestine-on-chip system is the TNO Intestinal Explant Barrier Chip (IEBC), a microfluidic organ-on-a-chip device developed to study the absorption and metabolism of pharmaceutical, biological and nutritional compounds across the intestinal wall [10]. This preclinical model aims to replicate the physiological characteristics of the human intestinal barrier, by reconstructing the structure and function in vitro. The in-house developed 3D design of the chip enables both apical (luminal) and basolateral (blood) fluid flow to the tissue explants. This medium throughput screening platform is mainly applied for drug absorption and metabolism studies. In addition, the IEBC allows us to study effects related to gut health, e.g., barrier integrity or the immune response, as well as host–microbe interaction studies [11].

The IEBC model is originally established with tissue explants of both human and porcine origin. Ex vivo intestinal tissue models reflect the complex physiology of the human intestinal tract and therefore serve as a highly realistic model for preclinical intestinal permeability studies [5,12]. The use of tissue is, however, associated with some practical issues: limited availability and complex logistics, limited viability and complexity in tissue handling. In addition, when using human tissue, there may be an underlying pathology involved for which patients receive surgery that might affect experimental outcomes. In addition, key characteristics of specific populations might not be represented, and the study design is impacted by the limited scalability. To overcome these issues, the IEBC model was adapted for cell monolayer culture instead of tissue explants. Traditionally, Caco-2 cell monolayers are used for epithelial barrier studies in vitro [13], but human tissue-derived enteroid monolayers are a promising, physiologically relevant and personalized alternative in vitro model to study drug transport [14] and metabolism [15]. 

The main aim of this research was to provide an experimental set-up and standardized read-outs for barrier integrity and permeability studies using the cell-on-chip system. The biological relevant functionality of Caco-2 and enteroid monolayers in this system is demonstrated, together with the impact of membrane characteristics on compound permeation.

## 2. Materials and Methods

### 2.1. Design and Fabrication of the System

The design and fabrication process of the general system is explained in our previous study [10] and is patented (US20230076661A1). A round insert for cell monolayer culture, the so-called disc, was designed with a cell growth area of 0.212 cm^2^. To allow the transfer of the disc from static culture to the chip, the IEBC was adapted with a twisting cap. The culture disc, fluidic chip and media reservoirs were 3D printed using a RapidShape S60 LED DLP printer (RapidShape GmbH, Heimsheim, Germany) and a biocompatible resin also used in dental applications, as previously published [16]. Preparation of the complete chip system is described below.

### 2.2. Caco-2 Cell Culture

Human colorectal adenocarcinoma cells (Caco-2) (ATCC, Manassas, VA, USA) were cultured in DMEM (Gibco, Thermo Fisher Scientific, Bleiswijk, The Netherlands) supplemented with 10% fetal bovine serum (Lonza, Geleen, The Netherlands), 1% non-essential aminoacids (Gibco, Thermo Fisher Scientific, Bleiswijk, The Netherlands) and 1% penicillin/streptomycin (Gibco, Thermo Fisher Scientific, Bleiswijk, The Netherlands). Cells were passaged upon reaching 80% confluency using TryplE (Gibco, Thermo Fisher Scientific, Bleiswijk, The Netherlands).

### 2.3. Human Enteroid Culture

Human proximal jejunum tissue (n = 3) was obtained from patients undergoing a pancreatoduodenectomy at Radboud University Medical Center (UMC), Nijmegen, The Netherlands. Since an underlying pathology was not impacting the intestine, this tissue was considered healthy. Following the Dutch Code of Conduct for Responsible Use, this leftover material could be used anonymously for research purposes without informed consent of the patients. The ethics committee of the RadboudUMC approved the use of tissue according to the Dutch Law on Human Research.

Three-dimensional enteroid cultures were established within 24 h after collection, according to established protocols [17,18] and as described by our lab recently [14]. Three-dimensional cultures were maintained in human IntestiCult™ Organoid Growth Medium (OGM) (StemCell Technologies, Vancouver, BC, Canada) supplemented with 1% penicillin/streptomycin (Gibco, Thermo Fisher Scientific, Bleiswijk, The Netherlands). 

### 2.4. Cell Monolayer Culture

Membranes cut out from standard Transwell^TM^ inserts (Corning Life Sciences BV, Amsterdam, The Netherlands) with low porosity (transparent) were used to create a support for the cells to grow in the cell disc. In addition, ipCELLCULTURE™ track-etched membrane filters (it4ip, Louvain-la-Neuve, Belgium) with low porosity (transparent) and high porosity (translucent) were also used as physical barriers. The specific characteristics of the membranes are provided in Table 1. A biocompatible sealing rubber (O-ring EPDM 70 compound 559270, ERIKS, Alkmaar, The Netherlands) was used to ensure accurate fixation of the membrane in the cell disc and to avoid technical leakage.

For Caco-2 monolayers, uncoated membranes were used. For enteroid monolayers, membranes were coated with 100 µg/mL collagen type I, rat tail (IBIDI) solution in acetic acid (0.01 *v*/*v*%, filtered) for 1 h at room temperature. After washing, these membranes were used to assemble the cell disc. The plastic parts for the cell disc were sterilized by UV and incubated in 20% anti-Biofilm solution (Ingenieurbüro Oetzel, Melsungen, Germany) for 1 h. After washing with phosphate-buffered saline (PBS, Sigma-Aldrich, Zwijndrecht, The Netherlands) and assembling the disc, the carrier with the disc was put in a sterile 12-well culture plate (Corning Life Sciences BV, Amsterdam, The Netherlands). 

Single cells were obtained from either Caco-2 or 3D enteroid cultures by TryplE treatment and seeded at a density of 1·10^5^ cells per membrane. Cultures were maintained in DMEM complete and human IntestiCult™ OGM for Caco-2 and enteroids, respectively. To prevent air bubble formation under the membrane, culture medium was put in the incubator (37 °C, 5% CO_2_) at least 1 h before use. When monolayers reached confluency, Caco-2 cells were transferred to an orbital shaker (25 rpm) in the incubator to increase differentiation by gentle shear stress [19]. Enteroid monolayers were differentiated by switching to human IntestiCult™ Organoid Differentiation Medium (ODM) (StemCell Technologies). 

### 2.5. Cell Monolayer Morphology

Brightfield images of cell monolayers during culture were obtained with an Olympus CKX41 (Evident Europe GmbH, Leiderdorp, The Netherlands) at 4× magnification. To visualize the viability of the cells, monolayers were incubated for 1 h at 37 °C with 0.5 mg/mL 3-(4,5-dimethylthiazol-2-yl)-2,5-diphenyltetrazolium bromide (MTT) (Sigma-Aldrich, Zwijndrecht, The Netherlands) in culture medium. Immunofluorescent staining was performed on whole mount monolayers, as described previously [14]. In short, monolayers were fixed for 1 h at 4 °C in 4% paraformaldehyde and, after a wash with PBS, incubated with blocking buffer (PBS containing 2% normal goat serum, 1% bovine serum albumin, 0.1% Triton X-100 and 0.05% Tween 20) for 1 h at room temperature to prevent non-specific binding. Next, monolayers were incubated with Alexa Fluor^TM^ Plus 555 Phalloidin (A30106, 1:100 Invitrogen, Thermo Fisher Scientific, Bleiswijk, The Netherlands) overnight at 4 °C. Nuclei were subsequently stained with DAPI, after which monolayers were mounted on a glass slide with ProLong^TM^ Gold Anti-fade Mountant (Invitrogen, Thermo Fisher Scientific, Bleiswijk, The Netherlands). Fluorescent images were obtained using an Axio observer microscope (Zeiss, Breda, The Netherlands) and processed with ZEN 2.3 pro (Zeiss, Breda, The Netherlands). 

To stain cross-sections of cell monolayers, paraformaldehyde-fixed enteroid monolayers were first embedded in HistoGel^TM^ (Epredia^TM^, Thermo Fisher Scientific, Bleiswijk, The Netherlands), after which they could be embedded in paraffin and sectioned. Slides were deparaffinized and rehydrated, followed by standard hematoxylin and eosin (HE) staining.

### 2.6. Chip System Preparation

To prepare the microfluidic system, the microfluidic chips were connected to medium reservoirs by Tygon and PTFE tubing (VWR). A 16-channel peristaltic pump (Ismatec, VWR, Amsterdam, The Netherlands) was used to obtain laminar flow through the system, with peristaltic tubing (Pharmed, VWR, The Netherlands). The end of the tubing could be connected either to the corresponding medium reservoir or waste, allowing either medium recirculation or flushing of the system, respectively. Tubing in the entire system was connected via microfluidic fittings (VWR for barbed connectors and Darwin microfluidics for metal connectors).

Before adding the discs with the cells, the microfluidic system was flushed with 70% EtOH and subsequently incubated with 20% anti-Biofilm solution (Ingenieurbüro Oetzel, Melsungen, Germany) for 1 h. After flushing with PBS, the complete chip system was run overnight at 37 °C with medium containing 1% bovine serum albumin (BSA) (Sigma-Aldrich, Zwijndrecht, The Netherlands), at a flow speed of 33 µL/min. This entirely saturates all parts of the system, thereby minimizing air bubble formation and non-specific binding of compounds of interest. 

To add the culture disc with cells into the system, it was transferred to a biosafety cabinet, the chips were opened from the top and the discs were added to the chip, after which they were closed using a twisting cap mechanism. Sufficient medium was added to the chip to ensure no air bubbles were trapped during this procedure. Flow speed was adjusted to 17 µL/min and the system was placed back into a cell culture incubator (37 °C, 5% CO_2_). After a short period of equilibration, transport medium (with/without compound of interest) was added to apical and basolateral reservoirs, with the basolateral medium containing 4% BSA. The medium volumes in the reservoirs depended on the duration of the experiment and the sample volumes required but were between 5 and 6.5 mL. To completely replace the medium inside the tubing and chip with fresh transport medium, the system was run without medium recirculation for 30 min prior to the start of measurements. Details on the permeability measurements are described below. 

### 2.7. Epithelial Barrier Integrity Measurements

To assess the cell monolayer integrity, a combination of 50 µM Dextran-FITC 4 kDa (FD4) (Sigma-Aldrich, Zwijndrecht, The Netherlands) and 50 µM Cascade-Blue 0.5 kDa (CB) (Sigma-Aldrich, Zwijndrecht, The Netherlands) was added to every incubation in the apical transport medium. Leakage to the basolateral medium was determined using the BioTek Synergy H1 microplate reader (BioSPX, Abcoude, The Netherlands). FD4 was measured with an excitation/emission wavelength of 485 nm and 528 nm, and CB was measured at 365 nm and 412 nm, respectively. Apparent permeability values (P_app_) were calculated according to Equation (1):(1)$$P_{app}=\left(\frac{dQ}{dt}\right)\times (\frac{1}{A\times C_{0}})$$
in which dQ/dt is the amount of drug transported over the barrier (ng/sec), A is the exposed area of the cell monolayer (cm^2^) and C_0_ is the initial dose concentration of the compound (ng/mL). All measurements were performed under flow (17 µL/min), and the chip system was inside a cell culture incubator (37 °C, 5%CO_2_) during the entire experiment, as described in Section 2.6. 

### 2.8. Epithelial Permeability Measurements

[^3^H]atenolol was used to measure paracellular transport, while [^14^C]antipyrine served as a reference marker for transcellular transport, at 10 and 1 kBq/mL, respectively. Both compounds were mixed with non-radiolabelled atenolol and antipyrine for a final concentration of 10 µM in the apical transport medium. Apical and basolateral medium reservoirs were sampled at t = 0, 1, 2 and 4 h. All measurements were performed under flow (17 µL/min), and the chip system was inside a cell culture incubator (37 °C, 5% CO_2_) during the entire experiment, as described in Section 2.6. The radioactive labelled compounds in the samples were detected in scintillation liquid (Ultima Gold, Perkin Elmer, Drachten, The Netherlands) using the Tri-Carb^®^ 5110 TR Liquid Scintillation Analyzer (Perkin Elmer, Drachten, The Netherlands). P_app_ values were calculated according to Equation (1), and subsequently, the transcellular-over-paracellular P_app_ ratio was calculated according to Equation (2): (2)$$A/A\;ratio=\frac{P_{app}Antipyrine}{P_{app}Atenolol}$$

### 2.9. Drug Cocktail Experiment

The drug cocktail designed to evaluate active drug transport and metabolism included 10 µM rosuvastatin (LGC Standards, OLY, UK), 50 µM metformin (Sigma-Aldrich), 10 µM midazolam (BUFA GmbH, Oldenburg, Germany) and 12 µM 7-hydroxycoumarin (umbelliferone, Sigma-Aldrich, Zwijndrecht, The Netherlands). For specific inhibition, 5 µM elacridar and 5 µM ketoconazole (both Sigma-Aldrich, Zwijndrecht, The Netherlands) were used. The drug cocktail was added to the apical medium reservoirs of chip experiments (flow) and the apical insert of static Transwell inserts. Inhibitors were added to both apical dose solution and basolateral transport buffer. During experiments, samples were collected at different timepoints from both apical and basolateral medium reservoirs, mixed with equal volume of ultrapure MeOH and stored at −20 °C until analysis by LC-MS/MS. 

For quantification, a standard of parental compounds and metabolites was prepared in the same medium as samples. Metabolites included were 1’-hydroxymidazolam (Sigma-Aldrich) and 7-hydroxycoumarin glucuronide (LGC Standards, OLY, UK). Both standard and samples were centrifuged for 10 min at 1500× *g* to precipitate protein, and subsequently, 1 µL was injected. Analysis was performed using a Shimadzu triple-quadrupole LCMS 8050 system with two Nexera XR LC-20AD pumps, a Nexera XR SIL-20AC autosampler, a CTO-20AC column oven and an FCV-20AH2 valve unit (Shimadzu, ‘s-Hertogenbosch, The Netherlands). The compounds were separated on a Synergi Polar-RP column (150 × 2.0 mm, 4 µm, 80 Å) with a 4 × 2 mm C18 guard column (Phenomenex, Woerden, The Netherlands). The mobile phase consisted of 0.1% (*v*/*v*) formic acid in ultrapure water (A) and 0.1% (*v*/*v*) formic acid in MeOH (B), and was set as 100% A (0–1 min), 100% to 5% A (1–8 min), 5% A (8–9 min), 5% to 100% A (9–9.5 min) and 100% A (9.5–12.5 min). The total run time was 12.5 min, and the flow rate was 0.2 mL/min. Peaks were integrated using the LabSolutions software version 5.96 (Shimadzu, ‘s-Hertogenbosch, The Netherlands). A limit of quantification (LOQ) was determined for each compound based on the standard curve and a limit of detection (LOD) was determined based on a noise signal. For each experiment, corresponding dose and blanks were included in measurements. Recovery was calculated as the percentage of original dose added to the system. 

### 2.10. Static Transwell Experiments

All static control experiments were performed in cell monolayers cultured on transparent Transwell inserts (Corning, Life Sciences BV, Amsterdam, The Netherlands). Procedures for seeding and culturing were exactly the same as previously reported [14]. Barrier integrity, permeability and drug cocktail experimental conditions were similar to as described above for cell monolayers on chip. The only difference was in the volume of apical (100 µL) and basolateral (600 µL) compartment, which was accounted for in P_app_ calculations. All measurements were performed inside a cell culture incubator (37 °C, 5% CO_2_).

## 3. Results

### 3.1. Cell Disc and Chip Design

The chip in this study comprised two parts: a movable culture disc that contains a porous membrane where cells can be grown on both sides, and a chip that contains the fluidic channels and houses the culture disc. This method provided the flexibility to culture under static conditions, characterize and differentiate cells in a traditional well plate and subsequently transfer them to a microfluidic chip without cell dissociation, to create a dynamic environment. A membrane that is required to support cell monolayer formation on the disc was fixed using a snap fit mechanism similar to IEBC [10] (Figure 1A). In static culture, the cells can attach and form a monolayer on the membrane (Figure 1B). Next, the disc was placed in a microfluidic chip from the top (Figure 1C). A cap then closed the chip and at the same time pushed down the disc to fix it. In this state, the disc and the cells interfaced the lower and upper channel, representing the apical and basolateral sides of the intestinal barrier. As shown in Figure 1D, and similar to IEBC, a peristaltic pump subsequently recirculated apical and basolateral media between the chips and reservoirs. The entire system is placed inside the incubator to perform all assays at 37 °C, 5% CO_2_.

### 3.2. Cell Monolayer Culture (Static)

To test whether the new cell disc design could support accurate cell monolayer formation, we started with membranes cut out of standard Transwell inserts to create a barrier in the disc. Both Caco-2 cells and enteroids were able to form monolayers during static culture, demonstrated by brightfield pictures (Figure 2A). To confirm complete growth area coverage and to evaluate cell viability, monolayers were stained by MTT conversion (Figure 2B). Remarkably, Caco-2 cells appeared to grow not only on the membrane but also to some extent on the plastic of the disc, while enteroid cells only covered the coated membrane. Immunofluorescent detection of actin showed accurate formation of cell junctions, with distinct differences in morphology between Caco-2 and enteroid monolayers (Figure 2C). Cells have a different shape and size, and cell density appeared to be higher in enteroid monolayers compared to Caco-2 monolayers. Cross-sections of the monolayers revealed that a single cell layer formed an epithelial barrier on the membrane (Figure 2D). Accurate polarization of the cells growing on Transwell membranes has been demonstrated by us elsewhere [14]. 

### 3.3. Epithelial Barrier Integrity

After growth and differentiation of the cell monolayers in static culture, the discs were transferred to the microfluidic chip system. We first needed to confirm epithelial barrier integrity under flow conditions. Traditionally, this is evaluated in Caco-2 monolayers by measuring the leakage of a fluorescent labelled dextran of 4 kDa (FD4). A monolayer is considered suitable for transport studies if the apparent permeability (P_app_) of this molecule is <1·10^−6^ cm/s [20], with a similar value identified for tissue in the IEBC system using tissue explants [10]. Interestingly, minimal-to-no FD4 leakage to the basolateral compartment was detected in all chips during the first 1–2 h of perfusion (Figure 3A), including the chips containing membranes without cells. There was no difference between Caco-2 and enteroids observed. Between 2 and 4 h of perfusion, P_app_ values for FD4 were still all below the cut-off value of 1·10^−6^ cm/s. In static culture, FD4 permeability can discriminate between membranes with and without cells (Appendix A). One chip without cells was identified as an outlier, with a P_app_ value of 0.74·10^−6^ cm/s, caused by disrupted laminar flow due to air bubble formation in the tubing (Figure 3A). These observations indicate that FD4 leakage in this system should be used to identify technical errors rather than epithelial barrier disruption.

To further examine the impact of the type of permeable membrane on the FD4 permeation, we next compared commercially available transparent and translucent membranes during microfluidic perfusion. The main difference between the membranes is the pore density (see Table 1 in the Section 2), while material and pore size were equal and thereby not visibly impacting cell monolayer formation. The permeability of FD4 was higher in chips with translucent membranes compared to chips with transparent membranes (Figure 3B). This effect was most pronounced for membranes without cells (2.7-fold difference) and enteroid monolayers (2.3-fold difference) and less obvious for Caco-2 monolayers (1.3-fold difference). Still, overall FD4 permeation was low and below established cut-off values and is, therefore, less biologically relevant.

In search of a more sensitive marker for barrier integrity, we next evaluated the leakage of a smaller molecule, Cascade Blue (CB, 0.5 kDa), to the basolateral compartment. In static culture, CB was able to permeate a membrane without cells (P_app_ between 16 and 24·10^−6^ cm/s), while permeation through an intact cell monolayer was low, with 0.5·10^−6^ cm/s and 0.08·10^−6^ cm/s for Caco-2 and enteroid monolayers, respectively (Appendix A). CB P_app_ values in the chip were significantly lower compared to static, with a very similar pattern to that observed for FD4 (Figure 3C). However, overall CB permeation was approximately 2-fold higher compared to FD4, allowing a more accurate read-out in this chip system. Again, the translucent membranes show higher P_app_ values compared to transparent membranes, irrespective of cell type used (Figure 3C).

Taken together, these results show that CB could be a more sensitive marker to assess the epithelial barrier integrity of a cell monolayer under flow conditions compared to FD4, and that there is a significant impact of the type of membrane used on compound permeability.

### 3.4. Epithelial Barrier Permeability

To further characterize the epithelial barrier created in the cell-on-chip system, we next tested the potential of the barrier to discriminate between low and high permeable compounds. Atenolol is a small molecule that passes the epithelium paracellular (Figure 4A) and has a fraction absorbed (Fa) of 50%. Antipyrine is a small molecule that translocates via the transcellular route and is highly permeable (Fa 100%). Thus, a functioning epithelial barrier shows 2-fold higher P_app_ values for antipyrine compared to atenolol when measured simultaneously.

P_app_ values for antipyrine measured in the chip system were similar for membranes with and without cells (Figure 4B), which was also observed in static monolayer cultures (Appendix A). Atenolol permeability in membranes without cells was higher compared to membranes with a cell monolayer (Figure 4C and Appendix A), showing the presence of functional tight junctions between the cells. P_app_ of atenolol measured in enteroid monolayers was similar to Caco-2. Again, an impact of the membrane used on compound permeability was observed, with distinct lower P_app_ values for both compounds in chips with transparent membranes compared to the chips with translucent membranes (Figure 4B,C). This membrane effect appeared to be clearer in Caco-2 monolayers than in enteroid monolayers. 

As expected, in the chips without cells, the A/A ratio detected was below the cut-off value of 2, since there was only a membrane separating apical and basolateral compartments and no functional cell barrier (Figure 4D). Similar observations were made in static Transwell insert controls (Appendix A). Caco-2 cell monolayers in the chip system showed an average A/A ratio of 2.9 on translucent membranes and 2.5 on transparent membranes (Figure 4D). For enteroid monolayers on translucent membranes, the average A/A ratio of 2.1 was above the cut-off value, while enteroid monolayers on the transparent membranes did not show an A/A ratio above 2 in the chip system (Figure 4D). In static culture on transparent Transwell membranes, the ratio was above the cut-off value (Appendix A), indicating accurate barrier functionality of the enteroid monolayers.

### 3.5. Drug Transport and Metabolism in Enteroid Monolayers

Next to passive transport, there is also active, carrier-mediated transport of compounds occurring in the intestinal epithelium. For example, active efflux of compounds back to the luminal (apical) side can significantly impact compound permeability. In addition, the epithelium can transform compounds in phase I and II metabolism. Both active transport and metabolism are important characteristics of the intestinal barrier, contributing to the first-pass effect, and therefore also need to be evaluated in the cell-on-chip model. 

We recently described the potential of enteroid monolayers for drug transport [14] and drug metabolism [15] in static conditions. Drug transporter expression and function in enteroid monolayers represented the biological variation present in vivo in human small intestinal tissue [14]. In addition, cytochrome P450 3A4 (CYP3A4) activity could be detected in enteroid monolayers, while being absent in Caco-2 monolayers [15]. Therefore, we focused on enteroid monolayers to measure active drug transport and metabolism in the cell-on-chip system. A drug cocktail was used to study both active transport and drug metabolism simultaneously [21]. The cocktail included rosuvastatin for breast cancer resistance protein (BCRP)-mediated transport, metformin for organic cation transporter (OCT)-mediated transport, midazolam for phase I metabolism by CYP3A4 and 7-hydroxycoumarin for phase II metabolism by glucuronosyltransferases (UGTs). In a second treatment group, the inhibitors elacridar and ketoconazole were added to the cocktail to show specificity. 

BCRP is an efflux transporter present at the brush border, limiting permeation of compounds from the apical to the basolateral side of the epithelium (Figure 5A). In both static (Figure 5B) and chip experiments (Figure 5 C,D), P_app_ values for rosuvastatin were low, suggesting the functional activity of BCRP. Specific inhibition of BCRP by elacridar can limit the efflux, thereby increasing permeation. Indeed, addition of the inhibitors increased the P_app_ value for rosuvastatin in static conditions (Figure 5B). This effect was less pronounced in the chip system with translucent membranes (Figure 5C) and surprisingly reversed in the chip system with transparent membranes (Figure 5D).

Metformin is actively transported by OCTs (Figure 5A), but at higher concentrations, permeation also occurs via diffusion [22]. Permeation of metformin in enteroid monolayers was higher compared to rosuvastatin, in both static (Figure 5B) and chip conditions (Figure 5C,D). Interestingly, in static conditions, the P_app_ value for metformin was lower compared to the P_app_ value in chip (Figure 5B,D). No selective inhibitor of OCTs was present in the drug cocktail applied. 

Midazolam and 7OH-coumarin were included in the drug cocktail to measure phase I and phase II metabolism, respectively (Figure 5A). In static Transwell cultures, enteroids display CYP3A4 activity as demonstrated by an increase in OH-midazolam over time (Figure 5E). This metabolite formation was absent in the presence of the CYP3A4 inhibitor ketoconazole (Figure 5E). Unfortunately, metabolite levels in the enteroid monolayers in the chip system were below the detection limit. The formation of 7-hydroxycoumarin glucuronide did confirm UGT activity in enteroid monolayers in both Transwell (Figure 5F) and chip systems (Figure 5G). 

## 4. Discussion

In this study, we described an intestine-on-chip model based on cell monolayers, as a new in vitro model for intestinal permeability studies. An established chip model for tissue explants (IEBC) was successfully adopted for cell culture and showed applicability with both Caco-2 cells and human enteroids for intestinal permeability studies. The main advantages of cell monolayers are the standardized experimental set-up, reproducibility and applicability for compound permeability studies. Being less complex compared to tissue, cell monolayers provide a suitable platform for the mechanistic understanding of absorption routes. 

The novelty of the specific design presented here is the cell culture outside of the chip. There are many different intestine-on-chip systems available, and the specific application of each system depends on the design, fabrication process, chip material and biological material used [23]. Several systems use cells in a chip to create an intestinal barrier [13,24,25,26,27]. These models all share the same principal in design: two channels separated by a porous membrane, with cells seeded in one channel. Injecting cells directly into small microfluidic channels in chips requires extensive training and reduces the reproducibility and reliability of the assay, as the cell injection strongly depends on the skillset of the operator. In addition, the low volume of most microfluidic chips requires frequent media refreshment, and therefore, long-term static culture is not preferred. The alternative is to use Transwell inserts and add these to a holder to apply flow [28], but this method increases the working volume and does not expose cells, especially on the apical side, to physiologically relevant shear stress. A major advantage of the newly developed cell-on-chip system described here is that the culturing of cells is started on a 3D-printed disc in a well-plate in static condition. The operator still has the option to let the cells attach and perform necessary assays in this condition, before transferring the cells to a fluidic chip. This method offers more flexibility to scientists to design a more reliable and reproducible organ on a chip assay, depending on their context of use. It is a step towards an easier-to-use system closer to current operating standards in cell culture labs. The complexity of intestine-on-chip systems is an advantage over simpler conventional in vitro models but makes the systems in general also complex to use.

Read-outs applied in this study included immunohistochemistry, fluorescent compounds, radioactive labelled drugs and LC-MS/MS, but the system allows us to include other measurements as well, e.g., oxygen or TEER sensors, or proteomics and genomics. An additional advantage of this chip system is that most parts are re-usable and, therefore, more sustainable compared to disposable chips. In the experimental set-up used here, culture media were re-circulated, allowing extended incubation time and increasing read-out sensitivity. But for other research questions, one-way flow is also an option in this modular system. Finally, the use of 3D printing as a fabrication technique also creates flexibility in prototyping any component of the cell-on-chip system. For example, the cell culture areas could be modified based on the context of use.

Both Caco-2 and enteroid monolayers showed similar results for the reference compounds included here. Yet, for specific research questions, there are likely different outcomes depending on the cell source. In general, P_app_ values measured in Caco-2 cell monolayers correlate well with in vivo Fa for passively transcellular transported compounds [29]. However, assessment of passive paracellular diffusion in Caco-2 cell monolayers is less accurate, mainly due to the much tighter and less permeable junctions between the cells compared to the native human intestinal epithelium [30,31]. Enteroid monolayers can classify absorption with a higher accuracy and show a higher precision to predict Fa values, especially for paracellular absorbed compounds [32]. In addition, the prediction of Fa for transporter-mediated drugs is low, because Caco-2 cells do not exactly display the same drug transporter patterns as in the human small intestine [33,34]. Also, their metabolic capacity is impaired compared to intestinal cells in vivo [35]. Enteroids have been shown to express drug transporters [32,36] and metabolizing enzymes [32,37] more similar to the tissue of origin and, therefore, provide a more accurate in vitro platform to study oral bioavailability.

Additional advantages of enteroids compared to Caco-2 cells are the potential to include biological variation, study regional differences in the intestinal tract and include specific patient populations. Including donor-to-donor variability allows for studying active transport for different polymorphisms, levels of expression and/or functionality. In the developmental phase, there was still a lot of variation in enteroid culture protocols, which highly affected study outcomes, but the standardized protocols available currently allow better comparison between different models. By using iPSC to generate enteroids, there is also an option to co-culture the epithelial cells with endothelial cells, immune cells and/or fibroblasts from the same donor. Building a more physiologically relevant intestinal barrier with not just the epithelium, but also stroma and endothelium, likely results in even more accurate permeability outcomes [38,39].

With drug exposure studies, it should be taken into account that compound absorption in the system can impair translatability [40]. In the static Transwell model, compound recovery ranged from 81% to 100% of the original dose (Appendix A). The recovery in the cell-on-chip system showed larger variation and was especially low for midazolam (Appendix A). To some extent, the recovery can be accounted for in the modelling of the data [41]; however, it can create issues with read-out sensitivity. For example, the recovery of midazolam was higher for the chips containing translucent membranes (41%) compared to the chips with transparent membranes (11%) (Appendix A) and might therefore explain why metabolite formation was only detected in the chip system with enteroid monolayers on translucent membranes. 

The laminar flow present in the chip had a clear impact on the permeability measurements. For FD4 and CB, only minor differences in P_app_ values were observed between chips with and without cell monolayers, while in static Transwell cultures, these differences were more evident. This suggests important technical differences between the systems. Measured antipyrine permeability could not discriminate between membranes with and without cells. This has been observed in other systems, where lucifer yellow [42] and antipyrine [43] permeability were also similar in microfluidic devices with and without cells. These results suggest that markers used as references in static models might be less applicable in OoC models, and further research is required to find optimal quality control parameters. 

Biologically, the laminar flow also had an impact on permeability measurements. For example, P_app_ values of antipyrine were lower in cells-on-chip compared to static cell monolayers, which has been observed previously [43]. In contrast, both atenolol and metformin permeability were higher in cells-on-chip compared to static conditions. Table 2 summarizes the findings of several studies examining the impact of flow on permeability measurements. There is a large variation in impact, and no clear pattern can be identified. The influence of flow on P_app_ values measured is likely dependent on the compound measured, i.e., dependent on the mechanism of transport involved. One important note is that cell monolayers in this study were cultured statically, and flow was only applied during permeability experiments. Applying flow during culture can result in improved cellular functions, since conditions are more physiologically relevant [19,44]. Interestingly, Caco-2 cells cultured under flow appeared to be more resistant to the bile salts in simulated intestinal fluid compared to static conditions [25]. Thus, while the impact of flow on permeability measurements still requires further investigation, it does create an environment that is more similar to in vivo. In addition, the basolateral flow allows connection to other organ systems in the future, e.g., liver-on-chip to mimic the first-pass effect of orally administered drugs. 

Most in vitro intestinal barrier models make use of a porous membrane to generate the physiological microenvironment. Considerations for the integration of membranes in organ-on-chip devices are extensively discussed by Corral-Nájera et al. [45]. The choice of membrane depends on the desired cell type and application, the membrane porosity, material and fabrication method. Here, we used track-etch membranes, which have a long history of use in barrier models, as they support cell attachment and monolayer growth, while creating a separation between two compartments [45]. A precise and homogenous pore size can be achieved by using this technique, allowing the exchange of drugs, nutrients and metabolites in a standardized setting. Transparent membranes are clear and light is able to pass through, which allows microscopical evaluation of the cell monolayers during culture. Translucent membranes only let some light through due to the crossed pore alignment and therefore appear white, making it difficult to detect cells with a microscope. As observed in this study, the specific characteristics of the membrane used can significantly affect the permeability of compounds in the barrier model. In general, the P_app_ value is never corrected for values measured in a no-cell control membrane. The impact of membrane permeability should be more acknowledged in in vitro permeability models, where the choice of membrane often depends solely on the optical characteristics of the membrane for practical reasons. Ideally, we will work towards a membrane-free chip system in the future, creating an even more physiologically relevant microenvironment. First attempts have been made to create a microfluidic device with a collagen-I-based membrane [46] or native extracellular matrix-based membrane [47]. However, issues with reproducibility in parameters, such as thickness and protein concentration, first need to be resolved before wide application in in vitro barrier models.

## 5. Conclusions

In conclusion, the cell-on-chip system presented here provides a versatile preclinical in vitro model for intestinal permeability studies. The model still requires validation with more compounds to reveal its full potential and translatability to in vivo intestinal permeability. In the future, intestinal cells-on-chip can also be connected through microfluidic perfusion to liver- and kidney-on-chip, to obtain a multi-organ-on-chip model. Ultimately, such a system can be applied in drug development to improve the prediction of human oral bioavailability. This is of particular interest for testing new biologicals with human-specific targets. However, making an in vitro model more complex also makes the conditions to take into account more complex, as demonstrated with the impact of flow and membrane permeability on experimental outcomes.

## 6. Patents

US20230076661A1 Microfluidic device for analyzing a membrane (https://patents.google.com/patent/US20230076661A1/en (accessed on 1 October 2024)).

## Figures and Tables

**Figure 1 micromachines-15-01464-f001:**
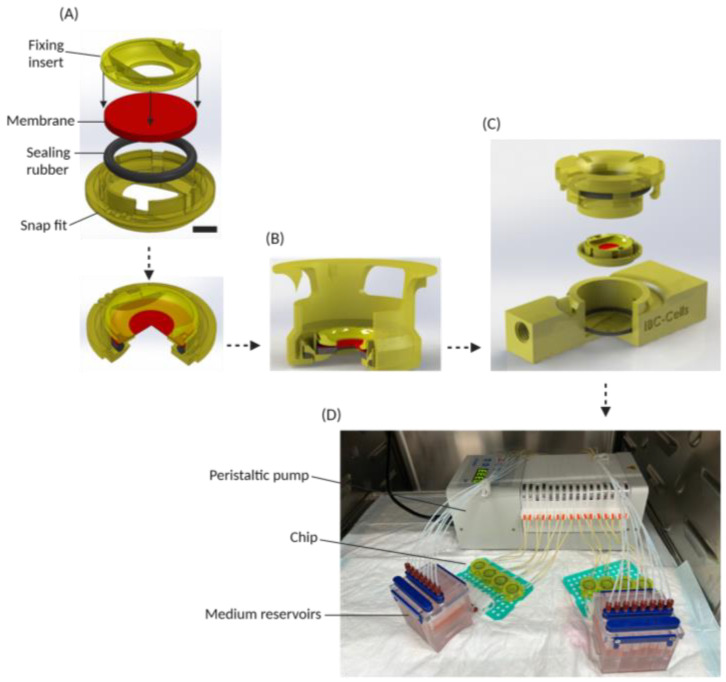
**Schematic overview of the cell-on-chip system**. (**A**) In-house developed 3D-printed disc with porous membrane and rubber ring for cell monolayer culture. Scale bar equals 2 mm. (**B**) Disc in a carrier for cell seeding and monolayer formation (static culture). (**C**) The disc with a cell monolayer can be transferred to a chip with dual flow. (**D**) Complete cell-on-chip system set-up, with eight chips connected to separate apical and basolateral medium reservoirs, with a peristaltic pump to create laminar flow, placed inside an incubator.

**Figure 2 micromachines-15-01464-f002:**
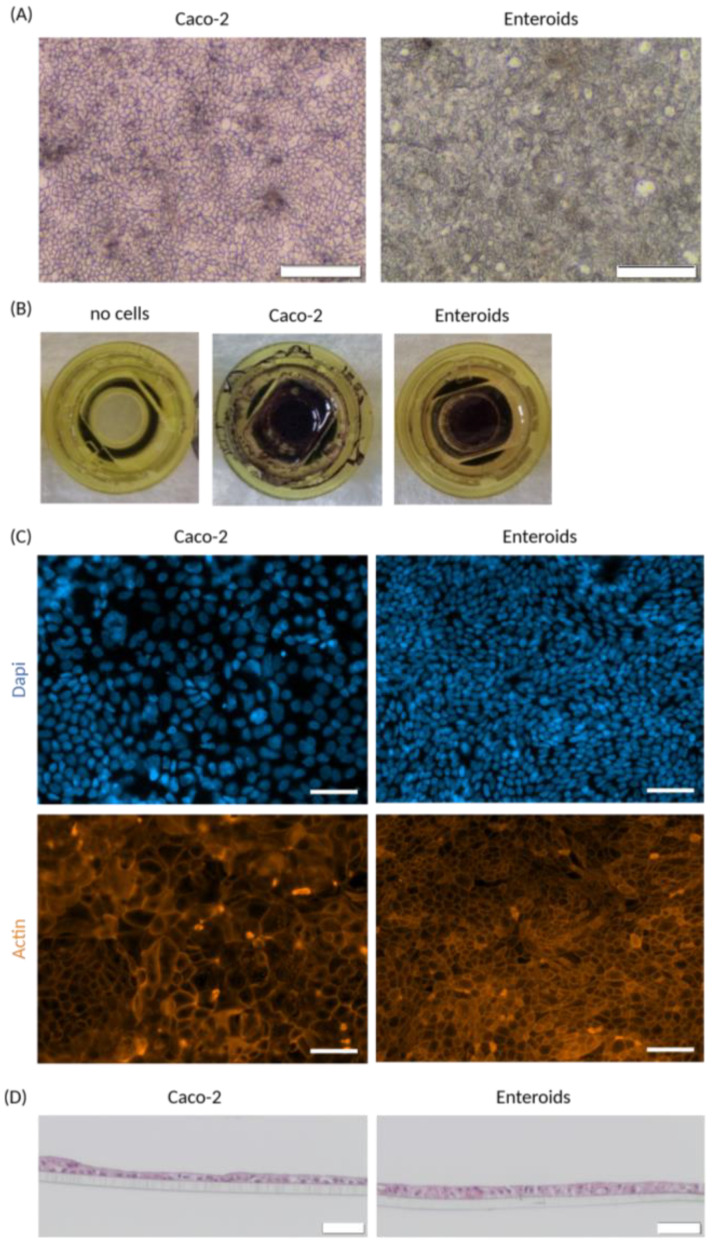
**Intestinal cell monolayer culture on disc**. (**A**) Brightfield images of epithelial cell monolayer cultures on a disc, derived from Caco-2 and enteroid cells. (**B**) Growth area coverage and monolayer viability indicated by MTT staining for Caco-2 and enteroid monolayers, compared to a disc with no cells. (**C**) Cell monolayer morphology assessed by whole-mount immunofluorescent detection of dapi (blue) and actin (orange). (**D**) HE staining of a cross-section of Caco-2 and enteroid monolayers, visualizing a single epithelial cell layer on the membrane. Scale bar equals 200 µm for (**A**) and 50 µm for (**C**,**D**).

**Figure 3 micromachines-15-01464-f003:**
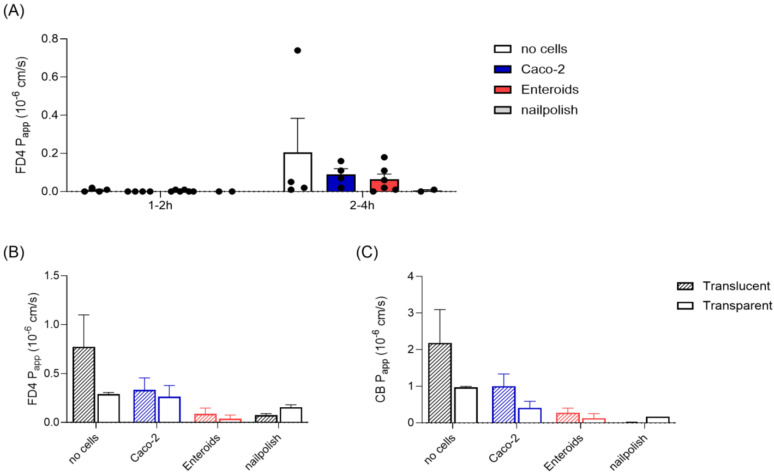
**Epithelial barrier integrity**. (**A**) Apparent permeability values (P_app_) for dextran-FITC 4 kDa (FD4) in cells-on-chip containing transparent Transwell membranes, determined between 1 and 2 h of perfusion and between 2 and 4 h of perfusion. Each dot represents a single chip. (**B**) P_app_ values for FD4 in cells-on-chip with translucent (dashed) and transparent (open) membranes, between 2 and 4 h of perfusion. (**C**) Cascade Blue 0.5 kDa (CB) permeability in cells-on-chip with translucent and transparent membranes, between 2 and 4 h of perfusion. n = 3–4 per group, with mean and SEM plotted.

**Figure 4 micromachines-15-01464-f004:**
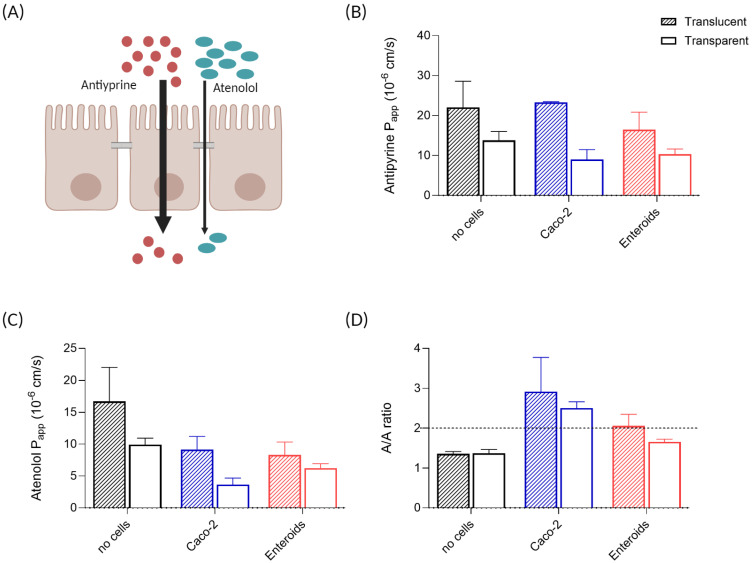
**Epithelial barrier permeability**. (**A**) Schematic overview of the transport routes for antipyrine (transcellular) and atenolol (paracellular) over the intestinal epithelial barrier. P_app_ values for (**B**) antipyrine and (**C**) atenolol in intestinal cell monolayers on chip, measured between 2 and 4 h of perfusion, for both Caco-2 and enteroid monolayers compared to no-cell control chip, with (**D**) corresponding antipyrine/atenolol ratios. Cut-off value for a functional intestinal epithelial barrier, i.e., an A/A ratio > 2, is indicated by a dashed line. For all graphs, the dashed bars represent monolayers on translucent membranes, and the open bars represent transparent membranes. n = 3–4 per group, with mean +/− SEM displayed.

**Figure 5 micromachines-15-01464-f005:**
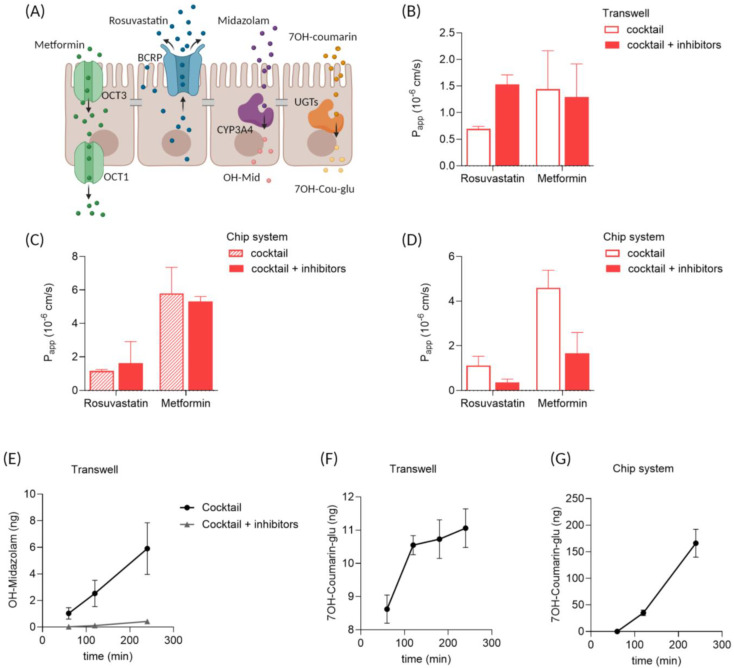
**Active drug transport and metabolism in enteroid monolayers**. (**A**) Schematic overview of the main routes for carrier-mediated active transport of rosuvastatin (BCRP) and metformin (OCT1) in the intestinal epithelial barrier, and phase I metabolism of midazolam (CYP3A4) and phase II metabolism of 7OH-coumarin (UGTs) by the epithelial cells. (**B**–**D**) P_app_ values for rosuvastatin and metformin in enteroid monolayer cultures, determined between 1 and 2 h of incubation, including inhibitors to block active transport, in (**B**) static transparent Transwell inserts, (**C**) chip system with translucent membranes and (**D**) chip system with transparent membranes. (**E**) OH-midazolam formation over time in enteroid monolayers in static Transwell cultures. (**F**,**G**) 7OH-coumarin-glu formation over time in enteroid monolayers in (**F**) static Transwell inserters and (**G**) chip system. The drug cocktail included metformin (50 µM), rosuvastatin (10 µM), midazolam (10 µM) and 7-hydroxycoumarin (12 µM). The inhibitors added were elacridar (5 µM) and ketoconazole (5 µM). n = 4–6 per group, with mean +/− SEM displayed.

**Table 1 micromachines-15-01464-t001:** Membrane characteristics.

	Company	Product nr.	Material	Pore Size Ø	Pore Density	Porosity	Pore Alignment	Membrane Thickness
Transwell^TM^ *transparent*	Corning^®^	#3470-Clear	Polyester (PET)	0.4 µm	4·10⁶ pores per cm^2^	0.5%	parallel	10 µm
ipCELLCULTURE™ *transparent*	It4ip ion track technology	2000M12/ 640N403/10	Polyester (PET)	0.4 µm	4·10⁶ pores per cm^2^	0.5%	parallel	12 µm
ipCELLCULTURE™ *translucent*	It4ip ion track technology	2000M12/ 811N403/10	Polyester (PET)	0.4 µm	1·10^8^ pores per cm^2^	12.6%	crossed	12 µm

**Table 2 micromachines-15-01464-t002:** **Impact of flow on permeability measurements.** Studies with a direct comparison between flow and static conditions on P_app_ values measured in cell monolayers.

	Cell Type	Compound	P_app_ A to B (×10^−6^ cm/s)	
			Flow	Static
This study *	Enteroids	FD4	0	0.01
		Atenolol	3.08	0.69
		Antipyrine	5.40	21.55
		Rosuvastatin	1.11	0.70
		Metformin	4.60	1.44
Zhang 2024 [13]	Caco-2	FD3	0.24	0.08
		Atenolol	0.72	0.26
		Antipyrine	32.9	38.5
Tan 2018 [24]	Caco-2	FD4	0.25	<0.1
Gleeson 2024 [25]	Caco-2	FD3	0.32	0.053
Sasaki 2022 [26]	Caco-2	Atenolol	0.10	0.43
Kulthong 2020 [43]	Caco-2	Antipyrine	5.4	22.7
Santbergen 2020 [27]	Caco-2/HT-29	Verapamil	19.5	20.9

* P_app_ values for transparent membranes, measured between 1 and 2 h.

## Data Availability

The data presented in this study are available on request from the corresponding author.

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
