# Peer review of "Intestinal Cells-on-Chip for Permeability Studies"

_micromachines, 2024, doi:10.3390/mi15121464_

Round 1
Reviewer 1 Report
Comments and Suggestions for Authors
In this manuscript, a new cell disc design was presented for accurate cell monolayer formation. A series of cell culture, epithelial barrier integrity and permeability, and drug transport tests were conducted for permeability studies. The structure is well-organized and the data is convincing. I think it should be accepted with just minor revisions as follows: some data about cell viability and growth rate should be added in “3.2. Cell monolayer culture (static) to validate the biocompatibility of this device.
Author Response
Comments 1: In this manuscript, a new cell disc design was presented for accurate cell monolayer formation. A series of cell culture, epithelial barrier integrity and permeability, and drug transport tests were conducted for permeability studies. The structure is well-organized and the data is convincing. I think it should be accepted with just minor revisions as follows: some data about cell viability and growth rate should be added in “3.2. Cell monolayer culture (static) to validate the biocompatibility of this device.
Response 1: We extensively tested all materials before application in the system, to ensure biocompatibility. The disc and chip were printed from a resin used in dental applications, which we now clarified in the methods section (line 98-99). In addition, a specific type of rubber was used for the sealing that was non-toxic to the cells. This information is now also added to the methods section: “A biocompatible sealing rubber (O-ring EPDM 70 compound 559270, ERIKS) was used to ensure accurate fixation of the membrane in the cell disc and to avoid technical leakage” (line 126-128).
In an attempt to further assess the viability of the cell monolayers cultured on the disc, we measured the cytosolic enzyme lactate dehydrogenase (LDH) in the culture supernatant using an LDH kit (Sigma-Aldrich). Unfortunately, results were inconclusive. Likely, the culture medium used was not compatible to measure LDH levels accurately with this kit, or the 24h incubation time was impacting LDH stability.
Still, the MTT staining shown in Figure 2B and the barrier properties measured subsequently indicate accurate cellular functions and therefore viability of the monolayers in the system. Therefore, we believe there is sufficient evidence that the system demonstrated here is biocompatible for both Caco-2 and human enteroid monolayer cultures.
Reviewer 2 Report
Comments and Suggestions for Authors
In this manuscript, the authors present an Intestinal Cells-on-Chip for epithelial permeability studies. A 3D-printed holder and a porous membrane was used to develop the cells-on-chip. Barrier integrity, permeability and drug transport were investigated using the models, The topic is relevant to the scientific community and matches the journal's purpose. However, there are several concerns, which should be addressed before it is considered for publication.
1. The future and challenge of the Intestinal Cells-on-Chip models are not summarized well. Tables should be provided to compare the different performance between this work and previous work.
2. The figures are rough and blurry. High resolution of images are required. The Figure 5 is not completely presented in the current manuscript. Scale bar is missing in Figure 1.
3. Relevant research background needs to be supplemented in introduction and discussion to illustrate the novelty of this work as compared to previous studies.
4. The methods for actin staining should be provided with more details such as the Triton X-100 concentrations.
5. A statistics section may be provided. Significance may be provided in Figure 3&4.
6. Please check the entire manuscript for typo errors such as line 256 and 261 “Error! Reference”.
Comments on the Quality of English LanguageThe English could be improved to more clearly express the research.
Author Response
We kindly thank the reviewer for the time taken to review the manuscript and to provide valuable suggestions to improve the manuscript. Please find below a point-by-point response.
Comments 1: The future and challenge of the Intestinal Cells-on-Chip models are not summarized well. Tables should be provided to compare the different performance between this work and previous work.
Response 1: Adding a complete table to compare the different performances of this work with other models is beyond the scope of this short communication. Besides, direct comparisons between different systems are usually impossible to make due to the large variation in system features. A summarizing Table on intestinal cells-on-chip models has already been published previously. We have added this information to the introduction, from line 57 it now states:
“Pimenta et al. provided a comprehensive overview of the main features and components of intestine-on-chip models reported in literature [9]. The large variation in fabrication techniques, materials used, architectural scaffold, fluid flow, mechanical stimulation, and the different applications of the models makes comparisons between studies challenging.”
In the discussion section, we do address the novelty of this system (from line 455) and we further compare features of this system with previous work (line 455 - 485). In the revised version of the manuscript we have added additional explanation, indicated by tracked changes.
Comments 2: The figures are rough and blurry. High resolution of images are required. The Figure 5 is not completely presented in the current manuscript. Scale bar is missing in Figure 1.
Response 2: Unfortunately, it appears that the figures lost some quality when converting to pdf for peer review. For final publication high quality images will be provided as separate files, enabling the editors to optimize final layout. In addition, some figures were improved in quality before resubmission, and a scale bar is added to Figure 1.
Comments 3: Relevant research background needs to be supplemented in introduction and discussion to illustrate the novelty of this work as compared to previous studies.
Response 3: We thank the reviewer for this suggestion. In the introduction, we provide background to indicate the relevance of adopting a previously published intestine-on-chip model for intestinal cell culture instead of tissue explants (from line 75):
“The use of tissue is, however, associated with some practical issues: limited availability and complex logistics, limited viability and complexity in tissue handling. Besides, when using human tissue, there may be underlying pathology involved for which patients receive surgery that might affect experimental outcomes. In addition, key characteristics of specific populations might not be represented and the study design is impacted by the limited scalability.”
In the discussion section, we address the novelty of this system (from line 455) and we further compare features of this system with previous work (line 455 - 485). In the revised version of the manuscript we have added additional explanation, indicated by tracked changes.
Comments 4: The methods for actin staining should be provided with more details such as the Triton X-100 concentrations.
Response 4: We have provided the details to methods section 2.5. It now states: “In short, monolayers were fixed for 1 hour at 4°C in 4% paraformaldehyde and, after a wash with PBS, incubated with blocking buffer (PBS containing 2% normal goat serum, 1% bovine serum albumin, 0.1% Triton X-100 and 0.05% Tween 20) for 1 hour at room temperature to prevent a-specific binding. Next, monolayers were incubated with Alexa FluorTM Plus 555 Phalloidin (Invitrogen, A30106, 1:100) overnight at 4°C”
Comments 5: A statistics section may be provided. Significance may be provided in Figure 3&4.
Response 5: In this study, we show data of technical replicates. To perform solid statistics with biological relevance, additional biological replicates need to be performed. There is a clear trend and the technical implications are clear from this data, and therefore we decided not to include a statistics section in this short communication.
Comments 6: Please check the entire manuscript for typo errors such as line 256 and 261 “Error! Reference”.
Response 6: Unfortunately, these errors were only present in the pdf version of the manuscript. In the original word document that was submitted, the correct references were displayed. We have checked the entire manuscript again for typo errors before re-submission.
Reviewer 3 Report
Comments and Suggestions for Authors
Recommendation: Major revisions necessary
Comments:
The paper presents an intestine-on-chip model to study the permeability of drugs in various conditions. The authors demonstrated detailed sections from design of the chip system then to the test of permeability of various barriers. This holds great value for the future research in the fields of invitro models and drug screening. The experiments are well-organized, and figures effectively communicate the findings. To ensure the advantages, understandability and logic of this paper, a major revision is required:
1. The flow was set as 0.017 uL/min. Please explain why the authors chose this parameter and add some references.
2. How does the presence of shear stress in the dynamic system influence cellular differentiation and transporter expression over long-term cultures?
3. Highlight the physiological relevance of flow in dynamic systems versus static systems more explicitly. The authors can consider discuss this by citing relevant studies, such as Doi: 10.1111/aor.13163; doi: 10.1039/C8LC00910D; 10.1088/1758-5090/ac2ef9;
4. In the introduction, please showcase the advantages of organ-on-chip devices by referring more studies in this area such as Doi: 10.1038/s43586-022-00118-6; Doi: 10.1002/adfm.202215043; Doi: 10.3389/fbioe.2022.840674.
5. There are many small mistakes in the manuscript. Please exam the whole manuscript and revise them. Below are some details:
A, in line 190, “A the exposed area” should be “A is the exposed area”. Similarly, C0 the initial dose concentration missed the “is”.
B, in line 156 and 261, the references number was wrong.
C, in line 294 and 295, “µM” should be “µm”. The caption of scale bar in Fig. 2D was missed.
D, The supplementary Figures in the manuscript should be removed because they have been presented in the Supplementary Information.
Author Response
We kindly thank the reviewer for the time taken to review the manuscript and to provide valuable suggestions to improve the manuscript. Please find below a point-by-point response.
Comments 1: The flow was set as 0.017 uL/min. Please explain why the authors chose this parameter and add some references.
Response 1: Unfortunately this was a mistake in the manuscript. The flow was set at 0.017 mL/min (=17 µl/min). We have corrected this in the manuscript.
Calculations on fluid flow and shear stress in this system were performed previously for the tissue explants (doi.org/10.1039/D1LC00669J). Flow applied for tissue was 2 mL/hour, corresponding to 0.033 mL/min. For the cells, besides shear stress, we also had to take practical implications into account, i.e. prevention of detachment of the cell monolayer and sufficient levels of media circulation for accurate measurement outcomes.
Comments 2: How does the presence of shear stress in the dynamic system influence cellular differentiation and transporter expression over long-term cultures?
Response 2: For this short communication, we only focused on the description of the new designed system and the impact of flow on permeability measurements. The cell monolayers were differentiated under static conditions, as indicated in the methods and results sections. The measurements in the dynamic system were relatively short-term experiments under flow (up to 4 hours).
We acknowledge that it is highly interesting, and relevant, to examine the impact of shear stress on cellular differentiation and transporter expression over long-term culture, however this was beyond the scope of this study.
Comments 3: Highlight the physiological relevance of flow in dynamic systems versus static systems more explicitly. The authors can consider discuss this by citing relevant studies, such as Doi: 10.1111/aor.13163; doi: 10.1039/C8LC00910D; 10.1088/1758-5090/ac2ef9;
Response 3: We highlight the physiological relevance of flow in the discussion section, between line 530-535: Applying flow during culture can result in improved cellular functions, since conditions are more physiologically relevant [16, 41]. Interestingly, Caco-2 cells cultured under flow appeared to be more resistant to the bile salts in simulated intestinal fluid compared to static conditions [33]. Thus, while the impact of flow on permeability measurements still requires further investigation, it does create an environment that is more similar to in vivo.
We have now added the relevant reference here [41, doi: 10.1111/aor.13163]
Comments 4: In the introduction, please showcase the advantages of organ-on-chip devices by referring more studies in this area such as Doi: 10.1038/s43586-022-00118-6; Doi: 10.1002/adfm.202215043; Doi: 10.3389/fbioe.2022.840674.
Response 4: We thank the reviewer for this suggestion, and have added the following information to the introduction (from line 54):
“These systems are able to mimic key aspects of human physiology by combining recent advances in tissue engineering and microfabrication [7]. The chips are designed to recapitulate the in vivo microenvironment of the cells, including chemical and mechanical stimuli, thereby maintaining tissue-specific functions [8].”
Comments 5: There are many small mistakes in the manuscript. Please exam the whole manuscript and revise them. Below are some details:
A) in line 190, “A the exposed area” should be “A is the exposed area”. Similarly, C0 the initial dose concentration missed the “is”.
Response 5A: Thank you for highlighting these mistakes. We have corrected this.
B) in line 156 and 261, the references number was wrong.
Response 5B: Unfortunately, these errors were only present in the pdf version of the manuscript. In the original word document that was submitted, the correct references were displayed. However, we have checked the entire manuscript again for these errors before re-submission.
C) in line 294 and 295, “µM” should be “µm”. The caption of scale bar in Fig. 2D was missed.
Response 5C: Thank you for highlighting these mistakes. The correct Figure legend now states: “ Scalebar equals 200 µm for (A) and 50 µm for (C) and (D).
D) The supplementary Figures in the manuscript should be removed because they have been presented in the Supplementary Information.
Response 5D: The reviewer is correct. For convenience during peer review, we provided the supplementary figures in the manuscript. For the revised version we have removed them from the main text and only provided the supplementary file.
Round 2
Reviewer 2 Report
Comments and Suggestions for Authors
I have no further comments.
Comments on the Quality of English LanguageThe English could be improved to more clearly express the research.
Reviewer 3 Report
Comments and Suggestions for Authors
The authors have canceled all my concerns. The current version can be accepted.